# Access Site Bleeding Complications with NOACs versus VKAs in Patients with Atrial Fibrillation Undergoing Cardiac Implantable Device Intervention

**DOI:** 10.3390/jcm11040986

**Published:** 2022-02-14

**Authors:** Enrico Guido Spinoni, Chiara Ghiglieno, Simona Costantino, Eleonora Battistini, Gabriele Dell’Era, Stefano Porcellini, Matteo Santagostino, Federica De Vecchi, Giulia Renda, Giuseppe Patti

**Affiliations:** 1Department of Translational Medicine, University of Eastern Piedmont, 28100 Novara, Italy; enrico.spinoni@gmail.com (E.G.S.); chiara.ghiglieno@gmail.com (C.G.); simona.costantino96@gmail.com (S.C.); eleonora.battistini@studenti.unipr.it (E.B.); 2Department of Thoracic, Heart and Vascular Diseases, Maggiore della Carità Hospital, 28100 Novara, Italy; gdellera@gmail.com (G.D.); stefanoporcellnimd@gmail.com (S.P.); matteo.santagostino@gmail.com (M.S.); federica.devecchi@gmail.com (F.D.V.); 3Department of Neuroscience, Imaging and Clinical Sciences, University G. D’Annunzio of Chieti-Pescara, 66100 Pescara, Italy; grenda@unich.it

**Keywords:** oral anticoagulation, pocket hematoma, cardiac implantable device, non-vitamin K antagonist anticoagulant, vitamin K antagonist

## Abstract

There are no data on procedure-related bleeding outcome with non-vitamin K antagonist anticoagulants (NOACs) versus vitamin K antagonist anticoagulants (VKAs) in patients with atrial fibrillation (AF) undergoing cardiac implantable electronic device (CIED) intervention. Our aim was to evaluate whether NOACs have a safety benefit even in terms of fewer hemorrhagic complications at the site of CIED implant. Consecutive AF patients receiving NOACs or VKAs at the time of CIED procedure were included in this observational, retrospective, and monocentric investigation. Primary endpoint was the incidence of post-intervention pocket hematoma. A total of 311 patients were enrolled, 146 on NOACs, and 165 on VKAs. The incidence of pocket hematoma was 3.4% in the NOAC versus 13.3% in the VKA group (*p* = 0.002). Primary outcome-free survival at 30-days was 96.6% in patients on NOACs and 86.0% in those on VKAs (*p* = 0.019). Multivariate analysis, adjusted by propensity-score calculation of inverse-probability-weighting, showed a significantly lower occurrence of pocket hematoma in patients receiving NOACs versus VKAs (HR 0.35, 95% CI 0.13–0.96, *p* = 0.042). Such NOACs benefit was confirmed versus patients on VKAs without peri-procedural bridging with low-molecular-weight heparin (HR 0.34, 95% CI 0.11–0.99, *p* = 0.048). The incidence of pocket infection, surgical pocket evacuation, ischemic events, and major bleeding complications at 30 days (secondary endpoints) was similar in the two groups. In conclusion, our data suggest that, among patients with AF undergoing implantable cardiac defibrillator or pacemaker intervention, the use of NOACs versus VKAs may be associated with significant reduction of post-procedural pocket hematoma, regardless of bridging with low-molecular-weight heparin in the VKA group. These results are hypothesis generating and need to be confirmed in a specific randomized study.

## 1. Introduction

Patients receiving a cardiac implantable electronic device (CIED) frequently present concomitant atrial fibrillation (AF) and therefore are often on oral anticoagulant therapy (OAC) for the prevention of thromboembolic events [1]. The number of patient candidates to CIED having AF is expected to increase in the future, as the progressive aging of the population worldwide enhances the prevalence of AF and enlarges the proportion of individuals with indication to pacemaker or implantable cardiac defibrillator (ICD) implant [2,3].

On the other hand, in patients undergoing CIED intervention, any anticoagulant treatment may increase the risk of procedure-related bleeding and pocket hematoma. This latter complication is relevant, as it predisposes patients to CIED infection [4,5,6]. In this setting, more recent evidence from clinical trials on patients receiving vitamin K antagonist anticoagulants (VKAs) showed no benefit on ischemic and bleeding events by a strategy of peri-procedural OAC interruption; of note, a significantly higher occurrence of hemorrhagic complications was observed in the subgroup undergoing VKA interruption and bridging with low molecular weight heparin (LMWH) [7,8]. Therefore, an uninterrupted VKA treatment is now recommended in AF patients undergoing CIED intervention [4,6].

Among patients with AF, the use of non-vitamin K antagonist anticoagulants (NOACs) instead of VKAs has been associated with at least similar protection from ischemic events and significant reduction of bleeding complications, especially intracranial hemorrhages [9,10,11,12,13]. Thus, NOAC utilization represents the treatment of choice in patients with AF to prevent stroke or systemic embolism (class I A) (1). Similarly to what suggested in patients on VKAs, an approach of non-interruption or minimal drug interruption is indicated in patients on NOACs receiving CIED procedures [4,6,14]. To date, there are no data on procedure-related bleeding outcome with NOACs versus VKAs in AF patients undergoing pacemaker or ICD procedures. Here we aimed at evaluating whether the use of NOACs may have a safety benefit over VKAs even in terms of fewer bleeding complications at the site of the device implant. 

## 2. Methods

This is an observational, real-world, retrospective, monocentric investigation, including consecutive patients on OAC for AF who underwent CIED intervention at Maggiore della Carità Hospital in Novara from January 2015 to March 2021. 

Specific inclusion criteria were: age >18 years; history of AF (paroxysmal, persistent or permanent) with CHA_2_DS_2_-VASc score ≥1; chronic therapy with a NOAC or VKA; indication to CIED procedure according to current guidelines [4]. Patients were included regardless of the type of CIED (implant, generator replacement, upgrading, or downgrading). Exclusion criteria were: moderate to severe mitral stenosis; presence of mechanical prosthetic valve; active bleeding or history of high bleeding risk; active infective endocarditis; major surgery in the previous month; platelet count <50,000 per microliter; ischemic stroke in the last 3 months or transient ischemic attack in the previous 3 days; fibrinolytic therapy in the previous 10 days; need for dual antiplatelet therapy (aspirin plus P_2_Y_12_ inhibitor); baseline hemoglobin <8 g/dL; severe hepatic disease with coagulopathy; serum creatinine value <15 mL/min or need for hemodialysis.

For all included patients, an electronic case report form was generated, where a unique pseudonymized code was assigned to each patient and individual data including patients’ demographic/clinical details, co-morbidities, medications, main laboratory test results, and procedural characteristics were collected. Management of NOAC/VKA treatment before and after the intervention was performed according to local practice patterns at the discretion of responsible physician. All CIED interventions were done by a trained staff with national certification, as recommended [4,6]. In the case of a new implant, surgical isolation of the left cephalic vein was performed; if cephalic vein was not available or it did not present anatomical characteristics suitable for device lead implant, the puncture of the axillary or subclavian vein was performed. Surgical pocket was set and sutured with single stitches on the ipo-dermal layer and then with continuous dermal absorbable stitch. In patients receiving a device generator replacement, a surgical incision was performed on top of the previous device pocket and the generator was replaced, with pocket remodeling-enlargement, if needed. After CIED intervention, all patients received a clinical assessment, also including the evaluation of the access site, the evening of the procedure and then once daily in the morning up to discharge. On the day after the procedure, the device was checked for adequate pacing/sensing measures (capture thresholds, sensing, and impedances) and chest X-ray was performed to confirm leads position and, in patients with subclavian access, to exclude pneumothorax. In the case of post-procedural pocket swelling, cryotherapy and/or compressive medication was provided, as appropriate. Patients were generally discharged on the subsequent day from the intervention, whether no complication occurred. After hospital discharge, a follow-up visit was planned at 7–10 days for stich removal and detection of early pocket suffusion/hematoma. A final follow-up visit was performed 30 days after the intervention in all patients. Approval by the Ethics Committee was not required as this is a retrospective study, did not use biological material, did not involve the collection, use, or transmittal of individual identifiable data and did not change standard practice patterns in all components of the study population.

Primary endpoint was the incidence of post-intervention pocket hematoma (up to 30 days), with comparison of this outcome measure between the two groups (NOAC versus VKA treatment at the time of the procedure). Pocket hematoma was defined by a palpable protruding mass (at least 2 cm of protrusion and radial length) [4]. Exploratory analyses were performed to compare different NOACs (dabigatran, apixaban, rivaroxaban, edoxaban) versus VKAs for the occurrence of the primary endpoint. 

Secondary endpoints were:Need for post-procedural surgical evacuation of pocket hematoma (e.g., in the case of tense, painful, extended hematoma causing compression of superficial perfusion, with subsequent high risk of erosion).Post-procedural pocket infection, defined as clinical presentation with inflammatory skin changes, including pain, swelling and redness, often associated with skin and soft tissue ulceration and drainage.Major bleeding at 30 days, defined as fatal or overt bleeding with a drop in hemoglobin level ≥23 g/dL, or requiring transfusion of at least 2 units packed blood cells, or hemorrhage into a critical anatomical site (e.g., intracranial, retroperitoneal) [15].Ischemic major adverse cardiovascular events (stroke, systemic embolism, transient ischemic attack, myocardial infarction) at 30 days.Length of in-hospital stay.

All outcome measure were adjudicated by direct source documentation. 

The normality of distribution of the parameters was assessed by Kolmogorov–Smirnov test. Continuous variables with normal distribution were described as mean ± standard deviation and compared by *t*-test, whereas variables with non-normal distribution were reported as median (interquartile range) and compared by Mann–Whitney U test. Categorical variables were indicated as number (percentage) and compared by Fisher’s exact test when the expected frequency was <5, otherwise the chi squared test (Yates’ corrected) was used. Event-free survival at 30 days in patients receiving NOACs versus VKAs was estimated by the Kaplan–Meier method and compared by log-rank test. 

Cox proportional hazard model was used to evaluate the association between covariates reported in Table 1 and Table 2 and the primary endpoint, with calculation of hazard ratio (HR) reported as hazard value and 95% confidence interval (CI). The final model of multivariate analysis included those covariates significatively associated with the primary endpoint at univariate analysis: age, gender, CHA_2_DS_2_-VASc score, HAS-BLED score, type of implanted device (ICD versus pacemaker implantation), and type of OAC (NOAC versus VKA). In order to minimize multiple biases due to the observational, retrospective study design, a propensity-score matching analysis was performed, initially including all available baseline characteristics. Eleven variables were then included in the model: age, gender, body weight, history of heart failure, serum creatinine, CHA_2_DS_2_-VASc score, HAS-BLED score, concomitant antiplatelet therapy, peripheral artery disease, previous myocardial infarction, and previous stroke (Appendix A). The predicted probabilities from the propensity-score model were used to calculate the stabilized inverse-probability-weighting (IPW) weight. HR for pocket hematoma by different OAC strategies (NOAC versus VKA) was then tested including the propensity-score calculation of IPW weight [16]. The threshold of statistical significance was 0.05 for all tests used (two-tailed). Statistical analyses were performed utilizing the STATA 14.0 software (StataCorp, LP, College Station, TX, USA).

## 3. Results

A total of 311 patients were consecutively included, 146 on NOACs at the time of the intervention (study group) and 165 on VKAs (control group). Main patients’ baseline characteristics are summarized in Table 1. Age was similar in the two groups, as well as body mass index, thromboembolic risk by CHA_2_DS_2_-VASC score, bleeding risk by HAS-BLED, prevalence of chronic renal failure and chronic obstructive pulmonary disease. Patients treated with NOACs more frequently were males, had paroxysmal AF and received pacemaker implantation. The rates of patients with ICD implantation, permanent/persistent AF, previous stroke or heart failure were higher in the VKAs group. The distribution of different NOAC types was 33.6% for dabigatran, 30.8% for apixaban, 26.7% for rivaroxaban, and 8.9% for edoxaban. The large majority of VKA patients were on warfarin (87.9%). Bridging with LMWH was performed in approximately one-third of patients, more frequently in those on VKAs. Details on procedural features are listed in Table 2. 

A total of 27 patients (8.7%) experienced a pocket hematoma at 30 days after the intervention. Main clinical and laboratory findings stratified by occurrence of this primary endpoint are indicated in Table 3. The incidence of pocket hematoma was 3.4% (*n* = 5) in the NOAC versus 13.3% (*n* = 22) in the VKA group (*p* = 0.002) (Table 4). At Kaplan–Meier analysis, primary outcome-free survival at 30-days was 96.6% in patients on NOACs and 86.0% in those on VKAs (*p* = 0.019) (Figure 1). Multivariate analysis showed that NOACs use was independently associated with lower risk of post-procedural pocket hematoma (HR versus VKAs use: 0.33, 95% CI 0.12–0.90, *p* = 0.032). Device type (e.g., ICD versus pacemaker implantation) and HAS-BLED score were predictors of a higher risk (Figure 2). Multivariate analysis, adjusted by propensity-score calculation of IPW, confirmed a significantly lower occurrence of pocket hematoma in patients receiving NOACs versus VKAs (HR 0.35, 95% CI 0.13–0.96, *p* = 0.042) (Figure 2). Among those patients who did not undergo bridging with LMWH (*n* = 253), the incidence of the primary endpoint was 3.4% with NOACs versus 9.9% with VKAs treatment (*p* = 0.03); here multivariate analysis confirmed the reduced risk of pocket hematoma in the NOACs group (HR 0.34, 95% CI 0.11–0.99, *p* = 0.048) (Figure 2). The incidence of pocket hematoma was lower for each NOAC type (dabigatran 3.1%; apixaban 0%; rivaroxaban 2.6%; edoxaban 7.7%) versus VKAs (13.3%). Multivariate analysis for the occurrence of the primary endpoint with different NOAC types compared to VKAs showed results consistent with those in the primary analysis on the overall population (Figure 3). 

The rates of other adverse events (secondary endpoints) are reported in Table 4. The occurrence of pocket infection, as well as of relevant, pocket-unrelated, major bleeding complications or ischemic cardiovascular events, was similar in patients treated with NOACs and VKAs. A strong trend towards lower need for post-intervention surgical pocket evacuation was observed in the former (0.7% versus 4.3%). Median in-hospital length of stay was 2 days (0–5) in the NOAC group and 2 days [1,2,3,4,5] in the VKA group (*p* = 0.38). 

## 4. Discussion

This observational study indicates that, among patients with AF undergoing CIED procedures, the use of NOACs versus VKAs is associated with significant reduction of post-intervention pocket hematoma. 

The rationale to compare NOACs and VKAs in the setting of CIED interventions is strong, as the greater safety profile with NOAC use, largely demonstrated in randomized and observational studies on the overall population of AF patients, was indeed not observed in various specific analyses on particular clinical settings. In fact, among patients with mechanical heart valves, the occurrence of both bleeding and ischemic events was higher in those receiving dabigatran versus VKAs, regardless of concomitant AF [17]. Moreover, in randomized studies on patients with AF the incidence of hemorrhagic complications, also including early post-procedural events, was similar with NOACs and VKAs after percutaneous ablation procedures [18], transcatheter aortic valve intervention [19], or cardioversion [20]. Finally, in a study-level meta-analysis on AF patients undergoing non-cardiac surgery, VKAs were as effective and safe as NOACs [21].

The hypothesis that NOACs may reduce access site bleeding complications after ICD or pacemaker implant has clinical relevance. In fact, the occurrence of pocket hematoma increases the risk of device infection (HR 7.7, as observed in the BRUISE CONTROL INFECTION study), with consequent poorer outcome over the long term [22]. In our investigation, patients on NOACs experienced lower rates of pocket hematoma than those on VKAs even if bridging with LMWH was not performed in the latter. This is important in as much as an approach without bridging must represent the standard strategy in patients on VKAs undergoing CIED interventions [4,6]. The greater safety of NOACs seems not related to a lower patients’ bleeding risk profile, as the two groups had similar HAS-BLED score, age and body mass index, as well as comparable prevalence hypertension, diabetes, liver disease, renal failure and concomitant antiplatelet treatment. This greater safety of NOACs compared with VKAs may be due to a shorter offset of action at the time of the procedure and a more stable anticoagulant effect in the post-operative period. Moreover, our data are strengthened by the results of multivariate analysis and propensity score matching, both accounting for potential confounding factors, especially for those variables with inequal distribution in the two groups. Notably, patients receiving VKAs at the time of the implant had INR values in the lower extreme of the therapeutic window (median 2.09); this does not justify per se a severe increase in bleeding propensity. In the NOAC group we also found a numerically lower incidence of post-intervention surgical pocket evacuation, whereas our study was underpowered for the occurrence of pocket infection complicating a pocket hematoma. Indeed, the reduction of access site bleeding events after CIED procedures by NOAC versus VKA use is consistent with what observed for other different bleeding types in randomized phase III trials and observational studies comparing such OAC strategies [1,9]. 

Our investigation has limitations inherent to observational and retrospective studies. Despite adjustments by multivariate analysis and propensity-score calculation of IPW, the risk of unmeasured residual confounding, as well as a higher baseline risk profile in patients receiving VKAs (e.g., higher ICD implants, greater percent in the high CHA_2_DS_2_-VASC and HAS-BLED range, higher prevalence of previous stroke and heart failure, more frequent bridging and anemia, lower platelet count), should be kept in mind. Furthermore, we aimed at including a monocenter cohort of consecutive patients, in whom individual data were accurately researched, but data collection on medical history and comorbidities was mainly based upon patient’s report, being therefore potentially biased. A possible treatment bias cannot be excluded, and the sample obtained might not be representative of the population intended to be analyzed. Finally, we were not able to perform a specific, robust analysis on the comparison between VKAs and different NOACs; however, an exploratory analysis on this topic yielded findings consistent with the main results in the overall population. 

In conclusion, NOAC use appears to be associated with a decreased risk of post-procedural pocket hematoma after CIED intervention compared to VKA use. These results are hypothesis generating and need to be confirmed in a specific randomized study. However, NOAC utilization overcomes intrinsic limitations of VKAs that are highly prevalent in the comorbid population of patients undergoing ICD or pacemaker implant. Thus, logical considerations and evidence-based data derived from our investigation both make NOACs the preferable anticoagulant drugs in this setting of patients. 

## Figures and Tables

**Figure 1 jcm-11-00986-f001:**
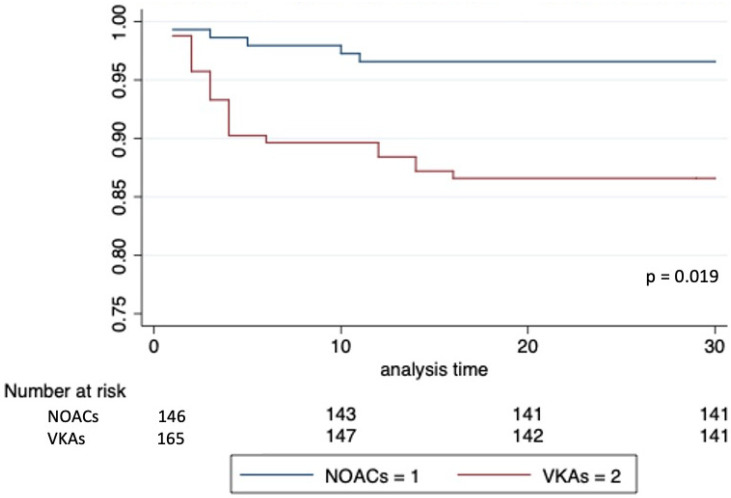
Kaplan–Meier curve for pocket hematoma-free survival at 30 days by different oral anticoagulation strategy (NOACs vs. VKAs). NOACs = Non-vitamin K antagonist anticoagulants; VKAs = Vitamin K antagonist anticoagulants.

**Figure 2 jcm-11-00986-f002:**
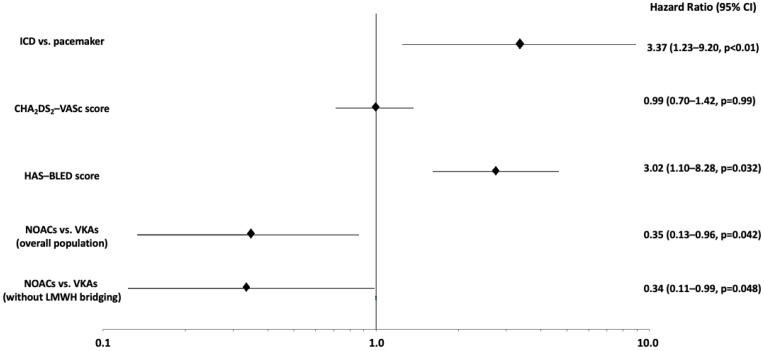
Hazard ratios for post-intervention pocket hematoma (primary endpoint) according to different oral anticoagulation strategies (overall population adjusted for IPW Propensity Score matching and patients without LMWH bridging), various device types, and CHA_2_DS_2_-VASc and HAS-BLED scores. ICD = Implantable cardiac defibrillator; IPW = Inverse-probability-weighting; LMWH = Low molecular weight heparin; NOACs = Non-vitamin K antagonist anticoagulants; VKAs = Vitamin K antagonist anticoagulants.

**Figure 3 jcm-11-00986-f003:**
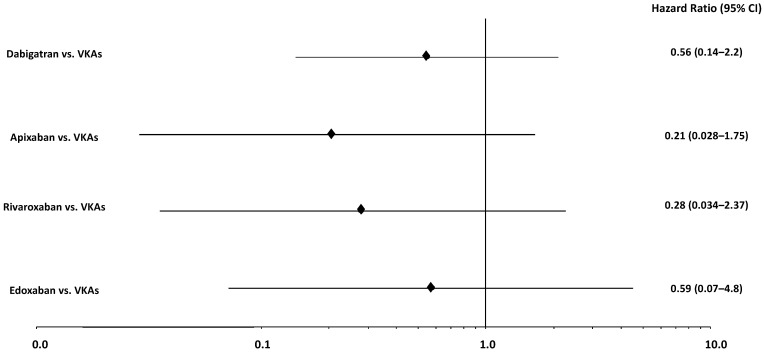
Hazard ratios for post-intervention pocket hematoma with different NOACs vs. VKAs. NOACs = Non-vitamin K antagonist anticoagulants; VKAs = Vitamin K antagonist anticoagulants.

**Table 1 jcm-11-00986-t001:** Main demographic/clinical and laboratory data at baseline in the included population.

Variables	NOACs	VKAs	*p* Value
	(*n* = 146)	(*n* = 165)	
Age (years)	78.4 ± 7.3	78.6 ± 9.5	0.84
Male gender	99 (67.8%)	94 (57.0%)	**0.049**
Body mass index (Kg/m^2^)	27.3 ± 5.0	26.3 ± 4.9	0.08
Device type			**0.020**
Pace-maker	120 (82.2%)	117 (70.9%)	
ICD	26 (17.8%)	48 (29.1%)	
AF type			**0.006**
Paroxysmal	35 (24.0%)	20 (12.1%)	
Persistent/permanent	111 (76.0%)	145 (87.9%)	
CHA_2_DS_2_-VASC score			0.40
1	3 (2.1%)	1 (0.6%)	
2	11 (7.5%)	9 (5.5%)	
3	27 (18.5%)	27 (16.4%)	
4	46 (31.5%)	47 (28.5%)	
5	41 (28.1%)	42 (25.5%)	
6	11 (7.5%)	22 (13.3%)	
7	6 (4.1%)	13 (7.9%)	
8	1 (0.7%)	3 (1.7%)	
9	-	1 (0.6%)	
Diabetes mellitus	38 (26.0%)	39 (23.6%)	0.53
Diabetes on insulin	21 (14.4%)	19 (11.5%)	0.45
Arterial hypertension	122 (83.6%)	132 (80.0%)	0.42
Peripheral artery disease	52 (35.6%)	63 (38.2%)	0.64
Previous MI	19 (13.0%)	28 (17.0%)	0.33
Previous stroke	11 (7.5%)	27 (16.4%)	**0.018**
Heart failure	76 (52.0%)	118 (71.5%)	**<0.001**
LVEF (%)	50.0 ± 13.3	45.2 ± 13.3	**0.002**
Chronic renal failure	52 (35.6%)	67 (40.6%)	0.37
COPD	21 (14.4%)	24 (14.5%)	0.97
HAS-BLED score			0.08
0	6 (4.1%)	8 (4.8%)	
1	96 (65.8%)	89 (54.0%)	
2	40 (27.4%)	52 (31.5%)	
3	4 (2.7%)	15 (9.1%)	
4	-	1 (0.6%)	
Liver disease	8 (5.5%)	11 (6.7%)	0.67
Previous major bleeding	12 (8.2%)	19 (11.5%)	0.33
OAC type			
Dabigatran	49 (33.6%)	-	
Rivaroxaban	39 (26.7%)	-	
Apixaban	45 (30.8%)	-	
Edoxaban	13 (8.9%)	-	
Warfarin	-	145 (87.9%)	
Acenocoumarin	-	20 (12.1%)	
Concomitant antiplatelet therapy	13 (8.9%)	14 (8.5%)	0.90
Peri-procedural LMWH bridging	4 (2.7%)	54 (32.7%)	**<0.001**
Hours from last OAC administration	35.1 ± 17.6	18.2 ± 10.6	**<0.001**
INR at the time of the procedure	-	2.09 (1.8–2.4)	
Serum creatinine (mg/dL)	1.1 ± 0.4	1.2 ± 0.5	0.15
eGFR (mL/min)	56.8 ± 17.5	56.1 ± 19.4	0.74
Hemoglobin (g/dL)	13.2 ± 1.9	12.8 ± 1.8	**0.042**
Platelet count (per microliter)	205,164.4 ± 60,194.5	191,606.1 ± 58,982.4	**0.046**

Values are expressed as *n*. (%) or mean ± standard deviation. Significant *p* values are indicated in bold. AF = Atrial fibrillation; COPD = Chronic obstructive pulmonary disease; eGFR = Estimated glomerular filtration rate; ICD = Implantable cardiac defibrillator; INR = International Normalized Ratio; LVEF = Left ventricular ejection fraction; LMWH = Low molecular weight heparin; MI = Myocardial infarction; NOACs = Non-vitamin K antagonist anticoagulants; OAC = Oral anticoagulant therapy; VKAs = Vitamin K antagonist anticoagulants.

**Table 2 jcm-11-00986-t002:** Procedural data.

Variables	NOACs	VKAs	*p* Value
*New implants*	*n* = 109	*n* = 82	
Device type			**0.033**
Pace-maker	89 (81.7%)	56 (68.3%)	
ICD	20 (18.3%)	26 (31.7%)	
Leads number			0.05
One lead	44 (40.4%)	47 (57.3%)	
Two leads	44 (40.4%)	21 (25.6%)	
CRT	21 (19.3%)	14 (17.1%)	
Venous access			0.09
Cephalic vein	60 (55.1%)	36 (43.9%)	
Subclavian vein	24 (22.0%)	30 (36.6%)	
Axillary vein	25 (22.9%)	16 (19.5%)	
Device side			0.52
Left	100 (91.7%)	73 (89.0%)	
Right	9 (8.3%)	9 (11.0%)	
*Generator replacement/downgrading*	*n* = 30	*n* = 73	
Device type			0.32
Pacemaker	27 (90.0%)	60 (82.2%)	
ICD	3 (10.0%)	13 (17.8%)	
Device side			0.06
Left	21 (70.0%)	36 (49.3%)	
Right	9 (30.0%)	37 (50.7%)	
*Upgrading*	*n* = 7	*n* = 10	
Device type			**0.036**
Pacemaker	4 (57.1%)	1 (10.0%)	
ICD	3 (42.9%)	9 (90.0%)	
Device side			**0.036**
Left	3 (42.9%)	9 (90.0%)	
Right	4 (57.1%)	1 (10.0%)	

Values are expressed as *n*. (%). Significant *p* values are indicated in bold. CRT = Cardiac resynchronization therapy; ICD = Implantable cardiac defibrillator; NOACs = Non-vitamin K antagonist anticoagulants; VKAs = Vitamin K antagonist anticoagulants.

**Table 3 jcm-11-00986-t003:** Main demographic/clinical and laboratory data in patients with vs. without post-procedural pocket hematoma.

Variables	Hematoma (*n* = 27)	No Hematoma (*n* = 284)	*p* Value
Age (years)	73.7 ± 11.4	78.9 ± 8.1	**0.002**
Male gender	17 (63.0%)	176 (62.0%)	0.92
Body mass index (Kg/m^2^)	27 ± 5.6	26.7 ± 4.9	0.77
Device type			**<0.001**
Pace-maker	10 (37.0%)	227 (80.0%)	
ICD	17 (63.0%)	57 (20.0%)	
AF type			0.14
Paroxysmal	2 (7.4%)	53 (18.7%)	
Persistent/permanent	25 (92.6%)	231 (1.3%)	
CHA_2_DS_2_-VASC score			**<0.001**
1	1 (3.7%)	3 (1.1%)	
2	1 (3.7%)	19 (6.7%)	
3	2 (7.4%)	52 (18.2%)	
4	8 (29.6%)	85 (29.9%)	
5	10 (37.0%)	73 (25.7%)	
6	2 (7.4%)	31 (10.9%)	
7	-	19 (6.7%)	
8	3 (11.2%)	1 (0.4%)	
9	-	1 (0.4%)	
Diabetes mellitus	11 (40.7%)	66 (23.2%)	0.12
Diabetes on insulin	5 (18.5%)	35 (12.3%)	0.36
Arterial hypertension	22 (81.5%)	232 (81.7%)	0.98
Peripheral artery disease	13 (48.1%)	102 (35.9%)	0.21
Previous MI	9 (33.3%)	38 (13.3%)	**0.006**
Previous stroke	7 (25.9%)	31 (10.9%)	**0.023**
Chronic heart failure	24 (88.9%)	170 (59.9%)	**0.003**
LVEF (%)	48.5 ± 13.2	36.7 ± 11.5	**<0.001**
Chronic renal failure	13 (48.1%)	106 (37.3%)	0.27
COPD	3 (11.1%)	42 (14.8%)	0.60
HAS-BLED score			**<0.001**
0	2 (7.4%)	12 (4.2%)	
1	8 (29.6%)	177 (62.3%)	
2	9 (33.3%)	83 (29.2%)	
3	7 (25.9%)	12 (4.2%)	
4	1 (3.7%)	-	
Liver disease	-	19 (6.7%)	0.17
Previous major bleeding	5 (18.5%)	26 (9.2%)	0.12
OAC type			**0.05**
Dabigatran	3 (11.1%)	46 (16.2%)	
Rivaroxaban	1 (3.7%)	38 (13.4%)	
Apixaban	0 (0)	45 (15.8%)	
Edoxaban	1 (3.7%)	12 (4.2%)	
Warfarin	19 (70.4%)	126 (44.4%)	
Acenocoumarin	3 (11.1%)	17 (6.0%)	
Concomitant antiplatelet therapy	7 (25.9%)	20 (7.0%)	**<0.001**
Peri-procedural LMWH bridging	11 (40.7%)	47 (16.5%)	**0.002**
Hours from last OAC administration	20.7 ± 12.1	26.7 ± 16.9	0.07
INR at the time of the procedure	2.3 ± 0.5	2.1 ± 0.7	0.47
Serum creatinine (mg/dL)	1.4 ± 0.8	1.1 ± 0.4	**<0.001**
eGFR (mL/min)	50.5 ± 20	57.0 ± 18.3	0.08
Hemoglobin (g/dL)	12.9 ± 1.6	13 ± 1.8	0.71
Platelet count (per microliter)	173,592.0 ± 48,069.1	200,288.7 ± 60,402.8	**0.027**

Values are expressed as *n*. (%) or mean ± standard deviation. Significant *p*-values are indicated in bold. AF = Atrial fibrillation; COPD = Chronic obstructive pulmonary disease; eGFR = Estimated glomerular filtration rate; ICD = Implantable cardiac defibrillator; INR = International Normalized; Left ventricular ejection fraction; LMWH = Low molecular weight heparin; MI = Myocardial infarction; NOACs = Non-vitamin K antagonist anticoagulants; OAC = Oral anticoagulant therapy; VKAs = Vitamin K antagonist anticoagulants.

**Table 4 jcm-11-00986-t004:** Study outcomes at 30-day follow-up.

Primary Endpoint	NOACs	VKAs	*p* Value
	(*n* = 146)	(*n* = 165)	
Pocket hematoma	5 (3.4%)	22 (13.3%)	0.002
**Secondary Endpoints**	**NOACs**	**VKAs**	***p* Value**
	**(*n* = 146)**	**(*n* = 165)**	
Need for surgical evacuation of pocket hematoma	1 (0.7%)	7 (4.3%)	0.06
Pocket infection	3 (1.8%)	2 (1.2%)	0.17
In-hospital length of stay	2 (1–5)	2 (0–5)	0.50
Non procedure-related major bleeding	1 (0.7%)	3 (1.8%)	0.37
Stroke/myocardial infarction/TIA/systemic embolism	0	0	NA

Values are expressed as *n*. (%) or median (interquartile range). NOACs = Non-vitamin K antagonist oral anticoagulants; TIA = transient ischemic attack; VKAs = Vitamin K antagonist anticoagulants.

## Data Availability

Not applicable.

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
