# Peer review of "Access Site Bleeding Complications with NOACs versus VKAs in Patients with Atrial Fibrillation Undergoing Cardiac Implantable Device Intervention"

_jcm, 2022, doi:10.3390/jcm11040986_

Round 1
Reviewer 1 Report
The study by Spinoni et al. provides interesting data on CIED pocket hematomas in patients on VKAs vs DOACs. This study has serious problems in that it is retrospective and that the two groups are different (either significantly or in some cases non-significantly) such that the outcomes could be easily challenged. The authors appropriately conclude that the results “appear” to suggest less bleeding and that the data is “hypothesis generating”. Such moderation should also be reflected in the concluding sentence in the abstract, which is all that many people might read.
I am concerned about the differences between the DOAC and VKA groups with most of those differences being associated with a potential increase risk for the VKA group. These include such things as higher ICD implants, greater percent in the high CHADSVASC range(6-8), more previous stroke, more heart failure, greater percent in the high HASBLED range(2-4), more bridging, most being fully anticoagulated during the procedure (INR 2.09), more anemia, lower platelet count. Essentially all of these characteristics are potential risk factors for more bleeding. I think the manuscript could be enhanced by a more extensive discussion of these limitations other than simply assuming that multivariate analysis and propensity matching accounts for or negates these differences.
Author Response
Reviewer n.1
The study by Spinoni et al. provides interesting data on CIED pocket hematomas in patients on VKAs vs DOACs. This study has serious problems in that it is retrospective and that the two groups are different (either significantly or in some cases non-significantly) such that the outcomes could be easily challenged. The authors appropriately conclude that the results “appear” to suggest less bleeding and that the data is “hypothesis generating”. Such moderation should also be reflected in the concluding sentence in the abstract, which is all that many people might read.
We thank the Reviewer for his/her comments.
- As requested, in the revised Abstract we have now added that: “In conclusion, our data suggest that, among patients with AF undergoing implantable cardiac defibrillator or pacemaker intervention, the use of NOACs versus VKAs may be associated with significant reduction of post-procedural pocket hematoma, regardless of bridging with low-molecular-weight heparin in the VKA group. These results are hypothesis generating and need to be confirmed in a specific randomized study.”
I am concerned about the differences between the DOAC and VKA groups with most of those differences being associated with a potential increased risk for the VKA group. These include such things as higher ICD implants, greater percent in the high CHADSVASC range (6-8), more previous stroke, more heart failure, greater percent in the high HASBLED range (2-4), more bridging, most being fully anticoagulated during the procedure (INR 2.09), more anemia, lower platelet count. Essentially all of these characteristics are potential risk factors for more bleeding. I think the manuscript could be enhanced by a more extensive discussion of these limitations other than simply assuming that multivariate analysis and propensity matching accounts for or negates these differences.
- Following the Reviewer’s indication, in the revised Discussion we have now added that: “Despite adjustments by multivariate analysis and propensity-score calculation of IPW, the risk of unmeasured residual confounding, as well as a higher baseline risk profile in patients receiving VKAs (e.g. higher ICD implants, greater percent in the high CHA2DS2-VASC and HAS-BLED range, higher prevalence of previous stroke and heart failure, more frequent bridging and anemia, lower platelet count), should be kept in mind.”
Reviewer 2 Report
Authors investigated bleeding complications with NOACs vs VKAs in patients with atrial fibrillation undergoing cardiac implantable device intervention. This is an interesting topic and may be the useful information, if the discussion goes a little deeper.
Comments are as follows;
- NOACs seems to be beneficial in reduction of bleeding complication. Line 240; Authors wrote as follows; “The greater safety of NOACs seems not related to a lower patients’ bleeding risk profile,. Please discuss the mechanism of this results. Why NOACs reduced hematoma compared to VKAs?
Minor points:
- The words “pace-maker” and “pacemaker” are used. It should be consistent to mean the same thing. It should be written as "pacemaker"
- Line 220, 227, authors use “vs” for example, “the use of NOACs vs VKAs is associated with significant reduction of post-intervention pocket hematoma.” What this vs mean? Authors should use versus instead of vs. Or, if authors meant to compare the effect in reduction of bleeding complication, directly describe like this. “the use of NOACs was associated with significant reduction of post-intervention pocket hematoma compared with VKAs.”
Author Response
Reviewer n.2
Authors investigated bleeding complications with NOACs vs VKAs in patients with atrial fibrillation undergoing cardiac implantable device intervention. This is an interesting topic and may be the useful information, if the discussion goes a little deeper.
We thank the Reviewer for his/her positive comments on our work.
Comments are as follows;
- NOACs seems to be beneficial in reduction of bleeding complication. Line 240; Authors wrote as follows; “The greater safety of NOACs seems not related to a lower patients’ bleeding risk profile. Please discuss the mechanism of this results. Why NOACs reduced hematoma compared to VKAs?
As requested, in the revised Discussion we have now included that: “This greater safety of NOACs compared with VKAs may be due to a shorter offset of action at the time of the procedure and a more stable anticoagulant effect in the post-operative period.”
Minor points:
- The words “pace-maker” and “pacemaker” are used. It should be consistent to mean the same thing. It should be written as "pacemaker"
We have modified the term “pace-maker” into “pacemaker” throughout the paper.
2. Line 220, 227, authors use “vs” for example, “the use of NOACs vs VKAs is associated with significant reduction of post-intervention pocket hematoma.” What this vs mean? Authors should use versus instead of vs. Or, if authors meant to compare the effect in reduction of bleeding complication, directly describe like this. “the use of NOACs was associated with significant reduction of post-intervention pocket hematoma compared with VKAs.”
We have modified the term “vs” into “versus” throughout the paper.
Round 2
Reviewer 1 Report
The authors have responded adequately to my comments